# Evaluation of CRP as a marker for bacterial infection and malaria in febrile children at the Douala Gyneco-Obstetric and Pediatric Hospital

**Yembu Ngwengi**[1☯¤a*], **Guy Pascal Ngaba**[1,2☯], **Martine Nida**[1,3☯¤b], **Dominique Enyama**[4,5☯]

**1** Faculty of Medicine and Pharmaceutical Sciences, University of Douala, Littoral, Cameroon, **2** Department of Biochemistry, Hematology and Microbiology, Douala Gyneco-Obstetric and Pediatric Hospital, Littoral, Cameroon, **3** Department of Infectious Diseases and Dermatology, Douala Gyneco-Obstetric and Pediatric Hospital, Littoral, Cameroon, **4** Faculty of Medicine and Pharmaceutical Sciences, University of Dschang, West, Cameroon, **5** Department of Pediatrics, Douala Gyneco-Obstetric and Pediatric Hospital, Littoral, Cameroon

☯ These authors contributed equally to this work.
¤a Current address: Department of Internal Medicine, Marie O Polyclinic, Douala, Littoral, Cameroon
¤b Current address: Department of Dermatology, Marie O Polyclinic, Douala, Littoral, Cameroon
* yemsy324@gmail.com

## Abstract

### Background

C reactive protein (CRP), a marker for the presence of inflammation, has been extensively studied for distinguishing bacterial from non-bacterial infection in febrile patients, but its role in excluding malaria in the febrile child has not been thoroughly evaluated.

### Method

This was a cross-sectional study at the Douala Gyneco-Obstetric and Pediatric Hospital which included all patients between the ages of one month and 16 years presenting with fever. Consenting patients received complete clinical examinations, then venous blood samples were collected and tested for CRP values, bacterial infection and malaria.

### Results

Samples of 220 children were analyzed. 142/220 had viral infections, 50/220 had malaria and 49/220 had bacterial infections. 7/220 had both malaria and bacterial infection. There was no significant difference between mean CRP values in malaria and bacterial infection (p = 1), but CRP means were significantly higher in malaria/bacterial infection than in viral infection (p<0.0001). Area Under the Receiver Operating Characteristics Curve (AUROC) values were 0.94 for malaria and 0.86 for bacterial infection, with a calculated cut-off of 23.6mg/L for malaria and 36.2mg/L for bacterial infection. At these cut-offs, CRP had a Positive Predictive Value (PPV) of 68.75% and 85.00% for malaria and bacterial infection respectively, with a Negative Predictive Value (NPV) of 94.74% and 89.05% respectively.

**Data availability statement:** The Data Availability Statement is under discussion and will be provided in a forthcoming update to this article.

**Funding:** The authors received no specific funding for this work.

**Competing interests:** The authors have declared that no competing interests exist.

## Conclusion

CRP can effectively exclude malaria and bacterial infection in febrile children in low-resource settings without the need for additional tests.

## Introduction

Fever is the most common reason that children and infants are brought to the emergency department, being responsible for as much as 20% of pediatric consultations [1]. Fever, also called pyrexia, is an elevated body temperature due to an increase in the body temperature's set point [2,3]. The upper limit of normal temperature most often used in pediatric populations is 37.8°C in the evening and 37.5°C in the morning [4,5]. Fever with Systemic Inflammatory Response Syndrome (SIRS) can result from a variety of conditions, with infections being the paramount cause in developing countries [6,7]. Many infectious agents can cause fever, including parasitic, bacterial, fungal and viral pathogens, which are difficult to distinguish from each other based on clinical presentation alone, especially in pediatric patients, in whom signs of severe infection are often subtle [7], and so most febrile children in low-resource settings receive antibiotics unnecessarily [8].

Malaria is a leading cause of fever and death in pediatric patients in developing countries, with a prevalence of 26.10% and a mortality of 3.8% in a recent study in Cameroon [9], and bacterial infections are a large determinant of pediatric mortality [10], with pneumonia being the leading cause of death (60% of deaths) in children under five years [11]. Many potential biomarkers have been evaluated to enable the prediction of bacterial infection in febrile patients, with C reactive protein (CRP) showing a lot of promise.

CRP is an acute-phase reactant that is produced by the liver whose concentrations rise during acute inflammation [12]. Many biomarkers commonly used in hospitals like leucocyte count (WBC- White Blood Cell count), absolute neutrophil count (ANC) and band cell count might aid in differentiating between causes of infection, but none have performed consistently enough to be used as standard point-of care tests [13–15]. CRP has been shown to be one of the only sufficiently accurate biomarkers that could safely and effectively reduce the prescribing of antibiotics; in a study in Beijing in 2015, CRP was as effective as procalcitonin (PCT) in predicting bacteremia, and more effective at predicting severe sepsis [16]; while in a study in Iran in 2017, it performed better than PCT and WBC at predicting bacterial meningitis [17]. A multicenter Swiss study in 2017 found CRP to be as effective as pneumococcal Polymerase Chain Reaction (p-PCR) at predicting bacterial pneumonia [18]. However, the majority of studies evaluating the diagnostic performance of CRP have taken place in high-resource settings.

Malaria has been shown to be a confounding factor for CRP analysis in developing countries. Several studies attest to the fact that it causes a significant rise in CRP values, and often with levels comparable to those seen in bacteremia; in Tanzania it outperformed WBC and ANC at predicting bacterial infection, but could not distinguish between bacterial infection and malaria on its own [19]. Independently of bacterial infection, CRP has been shown to accurately predict *Plasmodium* parasitemia in studies in Angola and Ghana [20,21]. Very little data are available on the subject in Cameroon.

It is therefore important to study the relationship between CRP, malaria and bacterial infection in detail so as to determine CRP's use in differentiating between these conditions, especially in pediatric populations where these two conditions are prevalent. Moreover, in

developing countries, an affordable and easily performed test will help improve management protocols for clinicians faced with a febrile child [22,23]. It would also tremendously lessen the financial burden on patients and caregivers, encouraging better compliance to treatment [24].

## Methods

### Study characteristics

This was a hospital based, cross-sectional analytic study, carried out from November 15th 2020 to May 15th 2021, for a duration of six months. This study was carried out at the Douala Gyneco-Obstetric and Pediatric Hospital (DGOPH). Douala is the capital city of the Littoral Region of Cameroon, a country in sub-Saharan Africa. It is the largest city and the economic capital of the country, with a population of more than 3.5 million inhabitants [25].

### Study population and sampling

The study targeted children presenting with fever at the Emergency Department (ED) or Out Patient Department (OPD) at DGOPH. Sampling was consecutive. Febrile children presenting at DGOPH were screened for eligibility. Inclusion criteria were: Children between the ages of one month to sixteen years (192 months) who presented with acute or subacute fever (T° > 37.1°C axillary- used for children above 2 years of age or rectal temperature > 37.6°C- used for children below two years of age, of ≤14 days duration) at the ED or OPD of DGOPH. Exclusion criteria included patients with neoplasm, inflammatory diseases, liver failure, and those who had suffered trauma or injury.

## Ethical considerations

The study was conducted only after ethical and administrative clearance had been obtained from both the University of Douala (No.2550 IEC-UD/04/2021/M) and the Review Board of DGOPH (No.0068 HGOPED/DG/CEI). All participants (or their parents/guardians) signed a consent form explaining the details of the study, including risks and benefits, before being included in the study. Autonomy was respected, as participants had the right to withdraw from the study at any time. Risks to participants were minimized to the best of our ability. The confidentiality of patients was maintained by using serial numbers rather than names on questionnaires. Privacy was maintained during all stages of interviews and physical examination of the patients. Samples were obtained following recommended guidelines and coded to ensure anonymity.

## Study procedure and laboratory analysis

Patients were approached at presentation at the ER or in the OPD. After obtaining informed consent, patients underwent a clinical interview, then received a complete physical exam in line with the Integrated Management of Childhood Illness (IMCI) recommendations by the WHO [26]. Clinical interview identified sociodemographic characteristics (age and sex), characterized the fever (degree and duration), and investigated past history. Topographical examination was done to search for infectious foci. The final diagnosis was based on clinical findings and results of investigations.

Specimen collection is described in detail in the full study protocol. About 2-5ml of venous blood was collected in sterile syringes and transferred to specialized aerobic and anaerobic Bact/Alert ® collection tubes for hemoculture analysis. About 2ml of venous blood was inserted into plastic dry tubes, to be used for immunoturbidimetric CRP analysis with Cobas C111®. About 1ml of blood was inserted into plastic EDTA tubes, to be used for thick smears.

A drop of blood (about 0.05ml) was used for an HIV antibody-based Rapid Diagnostic Test (RDT) and another drop for a SARS-COV-2 antibody-based RDT, as part of the hospital policy for all children presenting in the pediatric department. Where indicated, biological fluids from infectious foci were also collected.

Samples for CRP analysis were analyzed within two hours of collection, so no freezing was required. Samples were centrifuged in the BIOBASE ® centrifuge at a g-force of 1006 for five minutes, then plasma was used for immunoturbidimetric analysis with the Cobas C111®. All CRP assays were read independently by two operators, both blinded to the final diagnosis (at this stage neither smear or culture results were available.)

Blood for thick smears was stained with the Giemsa stain and examined under an electrically-powered optical microscope (Optika ®) at 100x magnification. Hemocultures were performed with the Bact/Alert ® detection system. All other cultures used the standard methods.

Only the principal investigator had results of the index test (CRP values) and reference tests (culture and thick smears). All other operators were blinded to both the final diagnosis and the results of the other tests. Besides the principal investigator, each test was read independently by two operators.

## Quality control and assessment

Laboratory investigations were carried out at the Douala Gyneco-Obstetric and Pediatric Hospital, Cameroon. Quality control was performed daily for each parameter before analysis of patient samples. All procedures were performed according to standard guidelines and instructions from product manufacturers.

## Statistical analysis and definitions

Data was entered into and analyzed with Epi Info Version 7. Sample size was calculated using Cochran's formula and a similar study [19]. Mahende and co. reported the prevalence of fever to be at 14.35% for pediatric consultations, which was the value we used to determine how many febrile children to sample. The resulting size was 196 children, though we eventually accumulated a larger sample. Out of the 250 children sampled, 220 had complete laboratory and clinical data. Quantitative variables were presented as means, standard deviation and medians where appropriate, or as frequencies and percentages after categorizing using predetermined or calculated cut-offs. Qualitative variables were presented as percentages and frequencies. CRP value, the outcome variable, was compared with each of the variables above. The means were compared with the two-sample t-test for means. CRP was considered positive if greater than 5mg/L, the cut-off used in all similar studies and recommended by the manufacturer [17–19]. CRP values less than 5mg/L, between 5-20mg/L, between 20-40mg/L, between 40-100mg/L and above 100mg/L were used to categorize patients with respect to the final diagnosis, values similar to those used in clinical decision-making, and which have been used in similar studies [19–21]. Thick smear positivity was defined by identification of trophozoites in any of the microscopic fields. Culture positivity was determined by growth in one or more inoculated media.

As per the STARD guidelines, Receiver Operating Curves (ROC) were plotted for CRP distribution, with the Area Under the Curve (AUROC) used to define diagnostic performance. Optimal CRP values were calculated from the ROC curves- this involved the use of Youden's J to select the best index from the plotted values, then sensitivity and specificity of CRP in both bacterial infection and malaria as well as positive and negative predictive values (PPV and NPV), were then calculated at the optimal cut-offs provided by the ROC curves.

Statistical significance was set at p < 0.05.

## Results

### Demographic and clinical characteristics

A total of 250 patients with fever were enrolled in the study. Thirty (12%) were excluded: twenty-four (9.6%) due to incomplete laboratory data, three (1.2%) due to neoplasm, two (0.8%) due to liver failure and one (0.4%) due to Kawasaki disease. The remaining 220 patients (88%) had complete clinical and laboratory data and no other likely source of fever other than infection. The median age was 32 months, the modal age range was 1–24 months (infants). Girls accounted for 115 (52.27%) of patients. The modal presenting temperature was 39°C, while the mean duration of fever was three days for viral infection and four days for bacterial infection and malaria. Twenty-one of the 220 children had mixed infections, while 199/220 had monoinfections with a single etiological agent- 29 children had only malaria, 42 had bacterial monoinfection and 128 had viral monoinfections. The most frequent diagnosis was viral upper respiratory tract infection, with 59 patients (26.81%), followed by viral gastroenteritis with 31 patients (14.09%). The most frequent bacterial infection was urinary tract infection with 10 patients (4.54%). 50 patients (22.72%) had malaria. Other clinical diagnoses are indicated in Table 1. Of the 220 blood cultures, 14 (6.3%) had growth of clinically significant organism, with the leading bacterium being *Klebsiella spp*. (28.57%). A total of 217 urine cultures were done and 11 (5.07%) were positive, with *Escherichia coli* as the predominant organism (72.73%). Stool cultures had a yield of 18.52%, with the leading bacterium being *Salmonella spp*. Pus cultures had a 100% yield, with the predominant organism being *Proteus mirabilis*. Among malaria patients, the minimum parasitemia was 15 Trophozoites of Plasmodium Falciparum per microliter (TPF/μl), while the maximum was 241,600 TPF/μl. The mean parasitemia was 39,638TPFμl/.

### CRP levels

The arithmetic means (with min and max values shown in parentheses) of CRP in malaria, bacterial infection and viral infection were 80.79mg/L (6.0–315.7), 84.41mg/L (0.23–324.0) and 17.82mg/L (0.2–237.36), respectively. Distribution of CRP levels according to patient diagnosis is shown in Table 2. No children with malaria had CRP levels less than 5mg/L. Most children with viral monoinfection (106/128) had CRP levels less than 20mg/L, while most children with bacterial monoinfection (30/42) and malaria monoinfection (23/29) had CRP levels greater than 20mg/L. The highest CRP levels were found in children with mixed bacterial infections and malaria, where all had CRP levels >40mg/L. Among patients with bacterial infection, CRP levels were highest in those with fever without source and lowest with skin infections. Among patients with viral infection, CRP levels were highest in those with COVID-19 and lowest in those with viral exanthems. Among malaria patients, CRP levels were independent of parasitemia. The degree of fever had no effect on CRP levels, but levels were lower in patients who presented after 7 days of fever than in those who presented before 7 days. Age had no effect on CRP levels, but levels were higher in females than males of almost every age group.

ROC analysis showed that CRP levels were positively correlated with positive cultures (Fig 1) and *Plasmodium* parasitemia (Fig 2). AUROC was 94.1% for malaria and 85.7% for bacterial infection. The optimal cut-off value of CRP was calculated to 36.2mg/L for bacterial infection and 23.6mg/L for malaria. Using these cut-offs, positive and negative fraction tables were drawn for malaria (Table 3) and bacterial infection (Table 4) which demonstrated that at these cut-offs, CRP had PPVs of 68.75% and 85.00% for malaria and bacterial infection respectively, with NPVs of 94.74% and 89.05% respectively.

**Table 1. Demographic and clinical characteristics of the study population.**

| Age (in months) | Number (N = 220) | Percentages (%) |
|---|---|---|
| 1–24 (infants) | 92 | 41.82 |
| 25–72 (preschoolers) | 82 | 37.27 |
| 73–144 (school-aged children) | 37 | 16.82 |
| 145–192 (adolescents) | 9 | 04.09 |
| **Sex** | **Number (N = 220)** | **Percentages (%)** |
| Male | 105 | 47.73 |
| Female | 115 | 52.27 |
| **Degree of Fever** | **Number (N = 220)** | **Percentages (%)** |
| 38–38.9oC | 67 | 30.45 |
| 39–39.9oC | 96 | 43.64 |
| ≥40oC | 57 | 25.91 |
| **Duration of Fever** | **Number (N = 220)** | **Percentages (%)** |
| <7 days (acute) | 194 | 88.18 |
| 7–14 days (subacute) | 26 | 11.82 |
| **Total Malaria** | **N = 50** | **100.00%** |
| **Total Bacterial Infection** | **N = 49** | **Percentages (%)** |
| Otitis Media | 3 | 6.12 |
| Tonsillitis | 8 | 16.32 |
| Rhinitis/Rhinopharyngitis/Rhinobronchitis | 2 | 4.08 |
| Gastroenteritis/Enteritis | 8 | 16.33 |
| Pneumonia/Superinfected Bronchiolitis | 7 | 14.29 |
| Meningitis/Enchephalitis | 2 | 4.08 |
| Urinary Tract Infection | 10 | 20.41 |
| Impetigo/Folliculitis/Abscess | 2 | 4.08 |
| Fever without source | 7 | 14.29 |
| **Total Viral Infection** | **N = 142** | **Percentages (%)** |
| Rhinitis/Rhinopharyngitis/Rhinobronchitis | 38 | 26.76 |
| Tonsillitis | 21 | 14.79 |
| Pneumonia/Bronchiolitis | 17 | 11.97 |
| Gastroenteritis/Enteritis | 31 | 21.83 |
| Meningitis/Encephalitis | 4 | 2.81 |
| Viral exanthem | 3 | 2.11 |
| COVID-19 | 4 | 2.81 |
| Fever without source | 24 | 16.90 |

## Discussion

This study described CRP levels among febrile children presenting at a tertiary hospital in urban Cameroon.

CRP means were comparable in bacterial infection and malaria, and in both of these conditions, levels were significantly higher than in viral infections (p<0.0001). Our study was unable to find a CRP value beyond which we could safely exclude either bacterial infection or malaria,

**Table 2. Distribution of CRP levels according to final diagnosis.**

| Final Diagnosis | CRP N (%) | | | | |
|---|---|---|---|---|---|
| | < 5mg/L | 5–20 mg/L | 21–40 mg/L | 41-100mg/L | > 100mg/L |
| Malaria monoinfection N = 29 | 00 (0.00) | 06 (20.69) | 07 (24.14) | 11 (37.93) | 5 (17.24) |
| Bacterial monoinfection N = 42 | 07 (16.67) | 05 (11.90) | 04 (9.52) | 17 (40.48) | 09 (21.43) |
| Viral monoinfection N = 128 | 66 (51.56) | 40 (31.25) | 17 (13.28) | 05 (3.91) | 00 (00.00) |
| Viral infection plus malaria N = 14 | 00 (00.00) | 00 (00.00) | 01 (7.14) | 09 (64.29) | 04 (28.57) |
| Bacterial infection plus malaria N = 7 | 00 (00.00) | 00 (00.00) | 00 (00.00) | 04 (57.14) | 03 (42.86) |

as the highest CRP values found in both conditions were very similar (both > 300mg/L). These results were similar to those of a similar study in Cambodia, where peak CRP values > 150mg/L were shared by both malaria and bacterial infection [27]. Few patients with malaria (6/50) and bacterial infection (12/49) had CRP levels < 20mg/L, the threshold most commonly used to eliminate viral causes with quantitative CRP analysis [19–21,27]. On the contrary, most viral monoinfections (106/128) had CRP < 20mg/L. Most patients with malaria (36/50) and bacterial infection (33/49) had CRP > 40mg/L. This is comparable to published data from

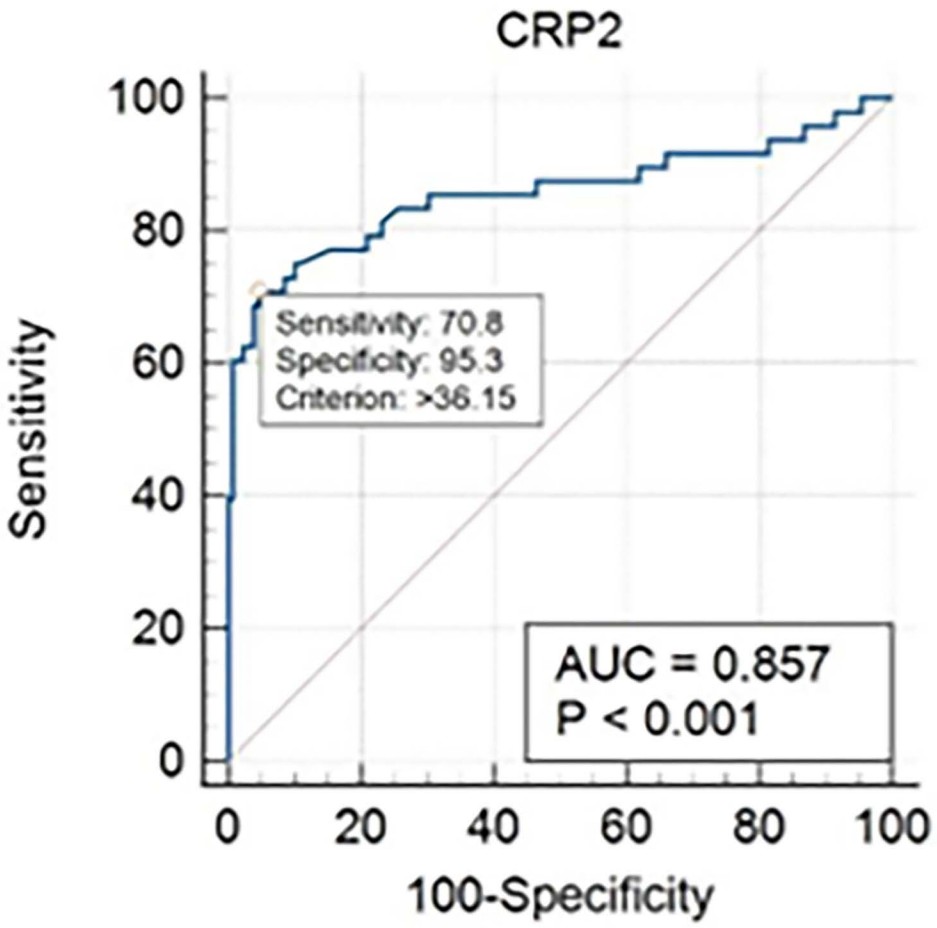

**Fig 1. Receiver operating characteristics curve for CRP in bacterial infection.**

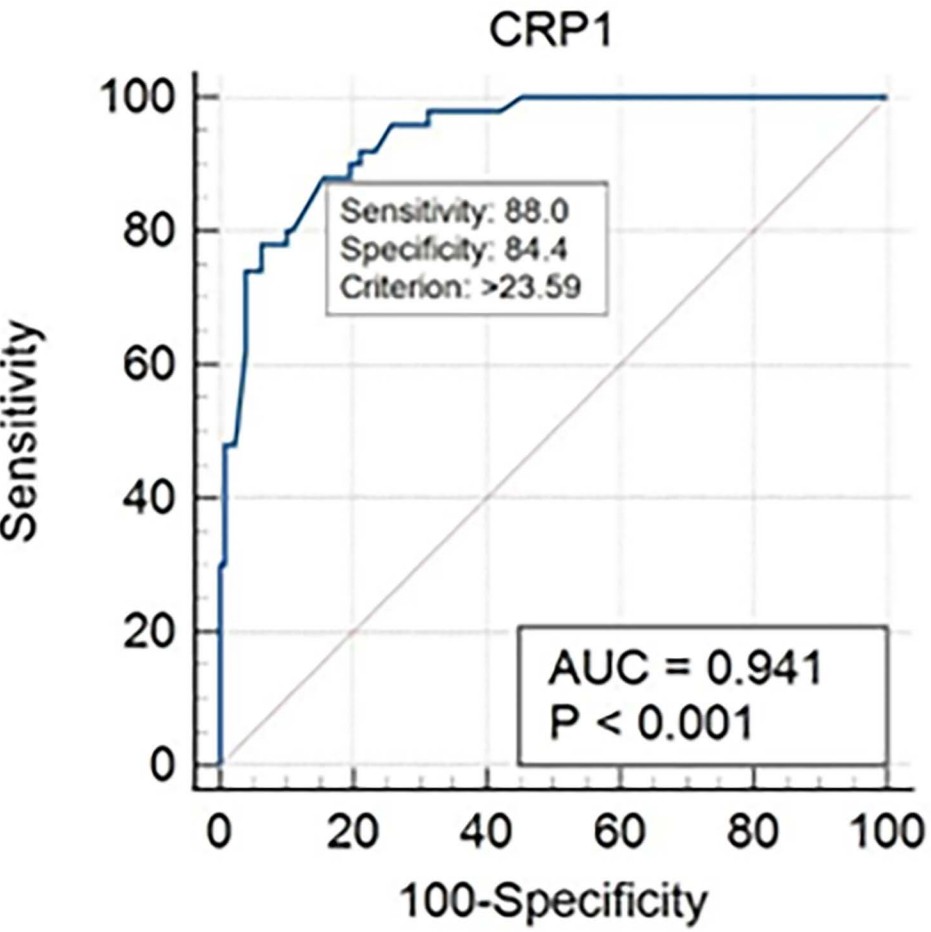

**Fig 2. Receiver operating characteristics for CRP and malaria.**

similar studies [19,27], suggesting that CRP is effective for distinguishing both malaria and bacterial infections from viral infections, but cannot separate malaria from bacterial infections in the febrile child on its own.

The available data indicates that CRP dosing has the potential to assist rational decision-making in febrile patients in lower-resource settings. AUROC values in our study were very similar to results from Cambodia and Tanzania (0.86 compared to 0.83 in both studies) [19,27]. These values are significantly higher (p<0.0001) than most other biomarkers used commonly to predict bacterial infection (PCT at 0.74, WBC at 0.56, ANC at 0.42) [19,27,28], suggesting that CRP may currently be the best biomarker for predicting bacterial infection in lower resource settings.

Based on the cut-off CRP values obtained by plotting Youden's J on the ROC curves, predictive values were calculated for malaria and bacterial infection. Positive predictive values (PPV) represent the proportion of true positives among the positive patients [29], as

**Table 3. Positive and negative values for CRP and malaria using optimal cut-off.**

| INFECTION | CRP <23.6 | CRP≥23.6 |
|---|---|---|
| MALARIA | 6 (FN) | 44 (TP) |
| VIRAL INFECTION | 108 (TN) | 20 (FP) |

**Table 4. Positive and negative values for CRP and bacterial infection using optimal cut-off.**

| INFECTION | CRP <36.2 | CRP≥36.2 |
|---|---|---|
| BACTERIAL INFECTION | 15 (FN) | 34 (TP) |
| VIRAL INFECTION | 122 (TN) | 6 (FP) |

defined by CRP > 23.6mg/L for malaria and CRP > 36.2mg/L for bacterial infection. PPV with this cut-off was 68.75% for malaria, lower than comparable PPV with the widely used histidine-rich protein malaria RDT, whose PPV was at 82.7% in a Rwandan study [30]. Both are inferior to the gold standard, thick smear. This suggests that CRP is less effective than RDTs in predicting malaria, and should not be used for this purpose. PPV for bacterial infection was at 85.00%, which is lower than the gold standard of bacterial culture. However, as mentioned above, CRP performs better in this regard than other alternatives like WBC count or ANC.

Negative predictive values (NPV) represent the proportion of true negatives among negative patients [29], as defined by CRP < 23.6mg/L for malaria and < 36.2mg/L for bacterial infection. NPV with this cut-off was 94.74% for malaria, a value higher than that of malaria RDTs in a similar study (85.4%), though still inferior to the gold standard. This suggests that CRP, at a cut-off of 23.8mg/L, performs better than malaria RDTs for excluding malaria in febrile children. NPV for bacterial infection with this cut-off was 89.05%. This is inferior to the gold standard of bacterial culture, but seemingly superior to other biomarkers currently in use for this purpose (60% for PCT) [27]. This suggests that CRP, at a cut-off of 32.6mg/L, might be the best available diagnostic test for excluding bacterial infection in febrile children in lower-resource settings.

It is important to note, however, that predictive values must be interpreted while taking the prevalence of the conditions in the target populations into consideration, as prevalence directly affects the predictive values. The most recent estimated prevalence of malaria in the pediatric population of Cameroon is about 26.10% [9], a value similar to the 22.7% found in this study. This implies that CRP can perform adequately for excluding malaria in this setting. However, many malaria-endemic areas have prevalence values lower (around 10% in both the Tanzanian and Cambodian studies used for comparison [19,27]) or higher (around 67% in the Rwandan study cited above [30]) than that used in this study, which would mean that comparing predictive values across these settings should be done cautiously. Nonetheless, the vast majority of sub-Saharan African countries have a prevalence between 10 and 50% [31], which means the values found in this study can be reasonably applied to most malaria-endemic countries in Africa.

Another important point to consider when analyzing the ROC curves and predictive values is that for clinical purposes, those with malaria and viral infections were considered in the malaria group, so only those with viral monoinfection were considered in the 'negative' group. Also, those with both malaria and bacterial infections were counted both in the malaria and bacterial infections groups, as statistically they both would test 'positive' for the various infection types. From the distribution of CRP levels shown in Table 1, it is evident that children with mixed infections generally had higher levels than children with monoinfection, with the more serious infection being the one predominantly affecting the CRP level. The viral monoinfection group, therefore, emerges as the best 'control' or 'negative' group, as the low values here correspond to the entity being excluded- that is, the definition of a negative case, a child with *neither* malaria nor bacterial infection. The ROC curves and predictive values thus gotten show that values below the optimum cut-off are likely to have *only* viral infections, and thus do not need antimalarials or antibiotics. Those with values above the cut-offs orient to *any* of the positive cases- that is, they could have malaria only, bacterial infection only,

malaria and bacterial infection, or malaria and viral infection. This is a large group, but once viral monoinfection is definitely excluded, distinguishing between them is fairly easy even in a low-income setting.

These values, taken globally, would suggest that a negative CRP (defined as <24mg/L for malaria and <36mg/l for bacterial infection), is usually correct in excluding malaria and bacterial infection in a febrile child. On the other hand, a positive CRP (defined as >24mg/L for malaria and >36mg/l for bacterial infection) is not enough to predict malaria in a febrile child on its own. Such a child would benefit from further evaluation with thick smears and malaria RDTs. However, if such a child tests negative for malaria (and confounding factors such as COVID-19 are excluded), then evidence suggests that there is a bacterial infection, and thus the patient would probably benefit from antibiotics.

## Limitations

It was a hospital-based, one center limited study and thus is not truly representative of the entire population of Douala, especially since DGOPH is a tertiary care center which is not accessible to a large proportion of the population. The study was limited to children presenting with fever, even though in some cases both bacterial infection and/or malaria present without fever. We were not able to exclude with 100% certainty all children who had received prior antibiotics or antimalarials, which could have resulted in false negatives with our cultures and thick smears. Evidence of viral infection could not be confirmed except in the cases of COVID-19. There may thus be a bias in the control group, which were considered to be viral infections because bacterial infection and malaria had been excluded.

## Research in context

There is a large body of evidence suggesting that CRP is an effective marker for distinguishing bacterial from non-bacterial infection, though most of these studies took place in high-resource settings (*Alcoba and co.*, Switzerland, 2017, and *Zhao and co.*, Beijing, 2020), and very few considered malaria (*Lubell and co.*, Cambodia, 2015 and *Mahende and co.*, Tanzania, 2017). To get the evidence needed, the authors searched the PubMed and Google Scholar databases, where search criteria included all studies from January 2010 to November 2020 which evaluated CRP's role in determining the type of infection in febrile patients, both pediatric and adult populations, and in all the regions of the world, though a focus was on lower-resource settings, that is, sub-Saharan Africa and Asia. The Pan African Medical Journal, the Plos One Journal and the BMC Infectious Diseases Journals were then searched for full articles. Search terms used included 'CRP use in febrile patients', 'CRP use to predict bacterial infection' and 'CRP use to predict malaria'. This evidence had been meta-analyzed by *Escadafal and co.* in 2020 in the BMJ Global Health Journal, with the pooled estimate being that CRP predicted malaria and bacterial infection with equal efficacy, but could not distinguish between both, and CRP was at least as effective as other common biomarkers like PCT, WBC count and ANC count in doing so. However, the authors admitted to widely different diagnostic performance across studies and the need for more thorough investigation in lower resource settings. This evidence is considered to be of very high quality, as all these studies were observational, thus eliminating recall bias; almost all the continents of the world were included in the analysis, eliminating regional bias; both adult and pediatric populations were considered, and the data had been previously meta-analyzed.

The authors feel that this study added the following contributions to the existing evidence:

- We correlated the degree and duration of fever to the final diagnosis and the level of inflammation, using CRP levels.

- We examined the prevalence of various viral and bacterial foci of infection in febrile children in a low-income setting, as well as *Plasmodium* parasitemia.

- We described bacterial species prevalence patterns in blood, stool, pus and urine cultures in febrile children in a low-income setting.

- We correlated CRP levels to a particular focus of bacterial or viral infection and *Plasmodium* parasitemia.

- We quantitatively compared maximum and minimum CRP thresholds in different infectious states in febrile children in a low resource setting.

Combined with the existing evidence, this study has the potential to provide clinicians in low-income settings with a rational protocol on management of febrile children in malaria-endemic settings. Such a protocol has been described in the full study (available in the library of the Faculty of Medicine and Pharmaceutical Sciences, University of Douala and the library of the Douala Gyneco-Obstetric and Pediatric Hospital). This protocol has the potential to reduce patient expenditure by reducing the need for expensive etiological tests, to reduce the over-prescription of antibiotics and to reduce misdiagnosis of febrile pediatric patients and thus reduce the infant mortality rate in low-resource settings.

## Author contributions

**Conceptualization:** Yembu Ngwengi, Guy Pascal Ngaba, Dominique Enyama.

**Data curation:** Yembu Ngwengi.

**Formal analysis:** Yembu Ngwengi, Guy Pascal Ngaba, Martine Nida.

**Funding acquisition:** Yembu Ngwengi.

**Investigation:** Yembu Ngwengi.

**Methodology:** Yembu Ngwengi, Guy Pascal Ngaba, Martine Nida, Dominique Enyama.

**Project administration:** Guy Pascal Ngaba, Martine Nida, Dominique Enyama.

**Resources:** Yembu Ngwengi, Guy Pascal Ngaba, Dominique Enyama.

**Software:** Yembu Ngwengi, Dominique Enyama.

**Supervision:** Guy Pascal Ngaba, Martine Nida, Dominique Enyama.

**Validation:** Guy Pascal Ngaba, Martine Nida, Dominique Enyama.

**Visualization:** Yembu Ngwengi, Martine Nida.

**Writing – original draft:** Yembu Ngwengi.

**Writing – review & editing:** Yembu Ngwengi.

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
