## [Decision Letter · Decision Letter 0]

4 Apr 2023

PONE-D-23-00167EVALUATION OF CRP AS A MARKER FOR MALARIA AND BACTERIAL INFECTION IN FEBRILE CHILDREN AT THE DOUALA GYNECO-OBSTETRIC AND PEDIATRIC HOSPITALPLOS ONE

Dear Dr. Ngwengi,

Thank you for submitting your manuscript to PLOS ONE. After careful consideration, we feel that it has merit but does not fully meet PLOS ONE’s publication criteria as it currently stands. Therefore, we invite you to submit a revised version of the manuscript that addresses the points raised during the review process.

I personally appreciate attempts to investigate the use of existing parameters for important applications under given resource conditions - particularly regarding diagnostics for malaria. The submitted manuscript has good potential to contribute to clarification of this significant issue. However, the study has some formal and methodological weaknesses which might limit the validity of the statements, especially regarding the selection of statistical parameters in different prevalence settings. The particular points of criticism by the reviewers are listed beyond. I would appreciate if you could revise the manuscript based on the reviewers´ comments.

We look forward to receiving your revised manuscript.

Kind regards,

Marc Reismann, MD, PhD

Academic Editor

PLOS ONE

Journal Requirements:

a) Did participants provide their written or verbal informed consent to participate in this study?

3. In your Data Availability statement, you have not specified where the minimal data set underlying the results described in your manuscript can be found. PLOS defines a study's minimal data set as the underlying data used to reach the conclusions drawn in the manuscript and any additional data required to replicate the reported study findings in their entirety. All PLOS journals require that the minimal data set be made fully available. For more information about our data policy, please see http://journals.plos.org/plosone/s/data-availability .

Upon re-submitting your revised manuscript, please upload your study’s minimal underlying data set as either Supporting Information files or to a stable, public repository and include the relevant URLs, DOIs, or accession numbers within your revised cover letter. For a list of acceptable repositories, please see http://journals.plos.org/plosone/s/data-availability#loc-recommended-repositories . Any potentially identifying patient information must be fully anonymized.

Important: If there are ethical or legal restrictions to sharing your data publicly, please explain these restrictions in detail. Please see our guidelines for more information on what we consider unacceptable restrictions to publicly sharing data: http://journals.plos.org/plosone/s/data-availability#loc-unacceptable-data-access-restrictions . Note that it is not acceptable for the authors to be the sole named individuals responsible for ensuring data access.

Additional Editor Comments:

Thank you for submission of the interesting manuscript. I personally appreciate attempts to investigate the use of existing parameters for important applications under given resource conditions - particularly regarding diagnostics for malaria. The submitted manuscript has good potential to contribute to clarification of this significant issue. However, the study has some formal and methodological weaknesses which might limit the validity of the statements, especially regarding the selection of statistical parameters in different prevalence settings. The particular points of criticism by the reviewers are listed beyond. I would appreciate if you could revise the manuscript based on the reviewers´ comments.

Reviewers' comments:

Reviewer's Responses to Questions

**Comments to the Author**

1. Is the manuscript technically sound, and do the data support the conclusions?

Reviewer #1: Yes

Reviewer #2: Partly

2. Has the statistical analysis been performed appropriately and rigorously? 

Reviewer #1: Yes

Reviewer #2: No

3. Have the authors made all data underlying the findings in their manuscript fully available?

Reviewer #1: Yes

Reviewer #2: No

4. Is the manuscript presented in an intelligible fashion and written in standard English?

Reviewer #1: No

Reviewer #2: Yes

5. Review Comments to the Author

Reviewer #1: I am honoured to have reviewed this article.

I have the following comments:

Throughout the paper, the decimal points have not been well placed making it difficult to appreciate data. Please check on the PDF format to appreciate this. This requires revision for appropriate interpretation.

In the main text, it is better not to start sentences with numerical eg 30. Better to have thirty for example.

In the abstract, it is documented that 220 were retained, once is unable to understand the meaning of this until the main results section has been read. Probably change to data was analysed for 220 participants or any other appropriate phrase.

The study period is indicated as 15th November to 15th May 2021, could the intention be 15th November 2020 to 15th May 2021?

In the Discussion, in interpreting the results, while it is true there appears to be greater value in NPV, the authors should choose the comparative words carefully, e.g 92.19% is not very close to 100% neither is 94.8 the same as 100%. Revision of the wording is suggested.

Reviewer #2: This study examines the levels of plasma C-reactive protein (CRP) levels among febrile paediatric patients, comparing among those with viral infection, bacterial infection and malaria (or combinations). It is concluded that low levels of CRP can effectively exclude malaria and bacterial infection and may be of use in resource-limited settings.

While this use of CRP to exclude malaria in such patients would appear to have potential in the setting described here, in which about 23% of the febrile children actually had malaria, the same predictive values would not apply in settings where the prevalence of malaria among febrile children was much different (higher). This issue should be pointed out in the manuscript.

Furthermore, making valid comparisons with the performance reported in other studies or using other markers needs to consider the prevalence of malaria among those study populations. Thus, the study in Rwanda using a malaria rapid diagnostic test was conducted on a group in which around 67% had malaria. Other things being equal, if that RDT had been conducted on the population with only 23% having malaria, the NPV would be around 97%, which is not dissimilar to (actually slightly higher than) the approximately 95% with CRP in the current study.

According to the literature, the prevalence of malaria among paediatric AFI cases in several malaria endemic countries is indeed around or slightly above 20%, but there are several settings in which the prevalence has been reported to be considerably higher than the 23% reported here. In these settings, low levels of CRP may not be as useful for excluding malaria as implied in this study.

Rather than making comparison of PPVs and NPVs across studies and markers without reference to the prevalence, estimating the markers’ performances using likelihood ratios (posterior odds/prior odds) would provide a fairer means of comparison. These could then be interpreted in terms of predictive values in different prevalence settings.

Other (minor) points:

Methods

• Study population and sampling: Was the duration of fever for inclusion actually less than 14 days or less than or equal to 14 days? (Compare with Table 1 “1-14 days”).

• Study Procedure and Laboratory Analysis: When stating the centrifuging values, it is better to state the relative centrifugal force (i.e., the “g” value) rather that the RPM. Without knowledge of the particular model of centrifuge used (the radius of rotation) the conditions are not explicit.

• Quality Control and Assessment: Was “…instruments…” intended to be “…instructions…”?

• Statistical Analysis and Definitions: The information on sample size calculation does not indicate the variables or measure(s) for which the calculation is being made.

• “Optimal CRP values were calculated from the ROC curves….” Could the definition of “optimum” be given? Was the calculation based on Youden’s J?

Results.

• Demographic and Clinical Characteristics: The intensity of parasitaemia is given in units of TPF/mm3. Assuming TPF refers to true positive fraction, the units do not seem to make sense. Should it be TP/mm3 (true positives per unit volume)?

• CRP levels: The CRP levels are given as geometric means, but what is the meaning of the values shown in parentheses following these geometric means? They do not appear to be distribution-based confidence intervals or min and max. They should be explained.

Discussion

• Third and fourth paragraphs: It is a tautology to state that the PPV or the NPV of the gold standard is 100%. They cannot be anything other.

• Fourth paragraph: Is the citation of reference [25] in the fourth paragraph correct?

6. PLOS authors have the option to publish the peer review history of their article (what does this mean? ). If published, this will include your full peer review and any attached files.

**Do you want your identity to be public for this peer review?** For information about this choice, including consent withdrawal, please see our Privacy Policy .

Reviewer #1: No

Reviewer #2: No

---

## [Author Response · Author response to Decision Letter 0]

1 May 2023

I have responded to these comments in a separate file labeled ‘Response to Reviewers’.

---

## [Decision Letter · Decision Letter 1]

23 May 2023

PONE-D-23-00167R1Evaluation of CRP as a marker for bacterial infection and malaria in febrile children at the Douala Gyneco-Obstetric and Pediatric Hospital.PLOS ONE

Dear Dr. Ngwengi,

Thank you for submitting your manuscript to PLOS ONE. After careful consideration, we feel that it has merit but does not fully meet PLOS ONE’s publication criteria as it currently stands. Therefore, we invite you to submit a revised version of the manuscript that addresses the points raised during the review process.

Most of the recent reviewer remarks have been addressed highly satisfactorily. One of the reviewers points at some particular statistical issues. I would be grateful if you could check your calculations and values based on the reviewers comments (see below).

We look forward to receiving your revised manuscript.

Kind regards,

Marc Reismann, MD, PhD

Academic Editor

PLOS ONE

Journal Requirements:

Reviewer's Responses to Questions

**Comments to the Author**

1. If the authors have adequately addressed your comments raised in a previous round of review and you feel that this manuscript is now acceptable for publication, you may indicate that here to bypass the “Comments to the Author” section, enter your conflict of interest statement in the “Confidential to Editor” section, and submit your "Accept" recommendation.

Reviewer #1: All comments have been addressed

Reviewer #2: (No Response)

2. Is the manuscript technically sound, and do the data support the conclusions?

Reviewer #1: Yes

Reviewer #2: Partly

3. Has the statistical analysis been performed appropriately and rigorously? 

Reviewer #1: Yes

Reviewer #2: No

4. Have the authors made all data underlying the findings in their manuscript fully available?

Reviewer #1: Yes

Reviewer #2: Yes

5. Is the manuscript presented in an intelligible fashion and written in standard English?

Reviewer #1: Yes

Reviewer #2: Yes

6. Review Comments to the Author

Reviewer #1: I do acknowledge, issues raised have been handled. I am grateful to have been part of the reviewers for this work.

Reviewer #2: Some of the issues that I raised on the previous draft of the manuscript have been addressed. However, there remain a few points that need further explanation and revision.

The first of these concerns the presentation of the sample size required for the study. The additional information is given that the prevalence of fever was expected to be 14.35%. It is not clear how this relates to the sample size calculation as all children in the study had at least some degree of fever. Furthermore, it is stated that the “resulting sample size (196 children) was assumed to give a 95% CI.” But a 95% CI of what measure and of what size? Could it be the CI of the area under the ROC curve, or, perhaps the CI around the value of expected positive predictive value for malaria or for bacterial infection after establishing an optimal cut-point, etc?

In the “CRP Levels” section of the Results, a revision has been made in which what were formerly referred to as geometric means, with possibly 25th and 75th percentiles in parentheses, are now called “mean levels (with 25th and 75% confidence intervals in parentheses) of CRP…” So, perhaps they are arithmetic means. However, it is still not certain what the values in parentheses are. I am fairly certain that the numbers in parentheses are not 25th and 75th confidence intervals. They appear possibly to be 25th and 75th percentile values, but normally these are given after presenting the median not the mean value. However, there is nothing intrinsically wrong in presenting the mean followed by 25th and 75th percentiles (i.e., the interquartile range) in parentheses, so long as the reader is adequately informed.

I am wondering if the number of children with malaria mono-infection who had CRP values greater the 20mg/L should be 23/29 rather than 23/26.

I am rather confused over the virtual summary tables (“virtual”, as they are not actually included in the manuscript) which contain the numbers of true and false positives (TP and FP) and false and true negatives (FN and TN) based on the optimal cut-points of CRP level for malaria and for bacterial infection. Given the values of PPV and NPV in the manuscript, I assume the numbers for the malaria table (TP, FP, FN and TN) would be 44, 25, 6 and 110. These numbers indicate a PPV of 63.77% and a NPV of 94.83% - almost the same as given in the manuscript. The sensitivity would then be 0.88 and the specificity 0.65. The vertical line on Figure 2 appears to be placed at too high a level of specificity.

For bacterial infection, I cannot satisfactorily work out what the numbers in the virtual table should be using the values of PPV and NPV given in the manuscript. The closest I can get is 39, 40, 11 and 130. These give a NPV of 92.19% (as reported) but a PPV of 49.37% (not 69.05% as reported in the manuscript). The sensitivity would then be 0.78 and the specificity 0.76. Again, the vertical line in Figure 1 does not exactly indicate this specificity. However, I may be wrong in my estimation of the numbers in the cells of the virtual tables.

The point of all this is that the reader might benefit from seeing the contents of these virtual tables.

I appreciate your explanation of TPF and apologize for my misunderstanding in the earlier review.

Minor point

Introduction, last sentence of 4th paragraph: “…data is available…” -> “…data are available…”

If these remaining issues can be satisfactorily addressed, the manuscript merits publication as it could make a valuable contribution to the clinical management of febrile children in Cameroon and other malaria-endemic countries.

7. PLOS authors have the option to publish the peer review history of their article (what does this mean? ). If published, this will include your full peer review and any attached files.

**Do you want your identity to be public for this peer review?** For information about this choice, including consent withdrawal, please see our Privacy Policy .

Reviewer #1: **Yes: ** Dr Justus Simba, MBChB, PhD.

Reviewer #2: No

---

## [Decision Letter · Decision Letter 2]

22 Jun 2023

PONE-D-23-00167R2Evaluation of CRP as a marker for bacterial infection and malaria in febrile children at the Douala Gyneco-Obstetric and Pediatric Hospital.PLOS ONE

Dear Dr. Ngwengi,

Thank you for submitting your manuscript to PLOS ONE. After careful consideration, we feel that it has merit but does not fully meet PLOS ONE’s publication criteria as it currently stands. Therefore, we invite you to submit a revised version of the manuscript that addresses the points raised during the review process.

In the reviews it is pointed towards a bias due to patient selection for the calculation of specificities and sensitivities affecting particularly ROC curves and tables 3 and 4. Double infections were not considered. Predictive values are probably affected. I subscribe in this point of view und would appreciate if you could revise the manuscript according to the reviewers´comments.

We look forward to receiving your revised manuscript.

Kind regards,

Marc Reismann, MD, PhD

Academic Editor

PLOS ONE

Journal Requirements:

Reviewers' comments:

Reviewer's Responses to Questions

**Comments to the Author**

1. If the authors have adequately addressed your comments raised in a previous round of review and you feel that this manuscript is now acceptable for publication, you may indicate that here to bypass the “Comments to the Author” section, enter your conflict of interest statement in the “Confidential to Editor” section, and submit your "Accept" recommendation.

Reviewer #1: All comments have been addressed

Reviewer #2: (No Response)

2. Is the manuscript technically sound, and do the data support the conclusions?

Reviewer #1: Yes

Reviewer #2: Partly

3. Has the statistical analysis been performed appropriately and rigorously? 

Reviewer #1: Yes

Reviewer #2: No

4. Have the authors made all data underlying the findings in their manuscript fully available?

Reviewer #1: Yes

Reviewer #2: No

5. Is the manuscript presented in an intelligible fashion and written in standard English?

Reviewer #1: Yes

Reviewer #2: Yes

6. Review Comments to the Author

Reviewer #1: My initial concerns were earlier addressed, the additional queries on data analysis by the other reviewer are well captured in the current manuscript.

Reviewer #2: I appreciate the revisions to the manuscript made by the authors, although I did not receive their detailed responses to my previous comments.

Some light on the issues with interpretation of the screening using CRP that I raised in my previous review has been shed by the 2-by-2 tables (Tables 3 and 4) now included in the R2- revised manuscript.

In order to use the concept of specificity the outcome variable must be categorized into “yes” and “no”. That is, if malaria is the outcome of interest, the two levels of outcome should be “yes” (referring to any case with malaria, whether malaria monoinfection, malaria plus viral infection or malaria plus bacterial infection) and “no” (referring to all cases without malaria infection, i.e., bacterial monoinfection and viral infection monoinfection). Doing so will yield different ROC curves and probably also different “optimum” cut-points for malaria and bacteria. Crucially, the predictive values, especially the positive predictive values, will be quite different from those reported in the manuscript. This is, of course, because when applying the CRP to screen for malaria, a positive value is true if the case actually has malaria but false if the case does not, i.e, could be viral infection or bacterial infection (it is not possible to identify those cases with only viral infection at the time of screening). A similar situation exists if one is trying to screen for bacterial infection (we should be comparing any case of bacterial infection (bacterial monoinfection or bacterial infection plus malaria) with cases not having any bacterial infection (which comprise viral monoinfection and malaria monoinfection).

As they stand, Tables 3 and 4 do not include all 220 cases and give a biased values for the specificity, PPV and NPV (and maybe sensitivity too).

It is recommended that the ROC curves be re-constructed for each of malaria and bacterial infection using the binary classifications of “yes” and “no” as indicated above, and thence the optimal cut-points, sensitivities, specificities and predictive values be re-determined. Tables 3 and 4 can then be re-constructed to include all 220 cases under study.

Don’t forget to include the units (mg/L) in the first sentence of the CRP Levels section of the Results.

Only a subset of the data has been included in the supplementary files.

7. PLOS authors have the option to publish the peer review history of their article (what does this mean? ). If published, this will include your full peer review and any attached files.

**Do you want your identity to be public for this peer review?** For information about this choice, including consent withdrawal, please see our Privacy Policy .

Reviewer #1: **Yes: ** Dr Justus M Simba, MBChB, PhD.

Reviewer #2: **Yes: ** Dr Alan Geater

---

## [Author Response · Author response to Decision Letter 2]

25 Jun 2023

The response to the reviewers can be found in the file named ‘Response to Reviewers 3’.

---

## [Decision Letter · Decision Letter 3]

10 Jul 2023

Evaluation of CRP as a marker for bacterial infection and malaria in febrile children at the Douala Gyneco-Obstetric and Pediatric Hospital.

PONE-D-23-00167R3

Dear Dr. Ngwengi,

We’re pleased to inform you that your manuscript has been judged scientifically suitable for publication and will be formally accepted for publication once it meets all outstanding technical requirements.

An invoice for payment will follow shortly after the formal acceptance. To ensure an efficient process, please log into Editorial Manager at http://www.editorialmanager.com/pone/ , click the 'Update My Information' link at the top of the page, and double check that your user information is up-to-date. If you have any billing related questions, please contact our Author Billing department directly at authorbilling@plos.org .

If your institution or institutions have a press office, please notify them about your upcoming paper to help maximize its impact. If they’ll be preparing press materials, please inform our press team as soon as possible -- no later than 48 hours after receiving the formal acceptance. Your manuscript will remain under strict press embargo until 2 pm Eastern Time on the date of publication. For more information, please contact onepress@plos.org .

Kind regards,

Marc Reismann, MD, PhD

Academic Editor

PLOS ONE

Reviewers' comments:

Reviewer's Responses to Questions

**Comments to the Author**

1. If the authors have adequately addressed your comments raised in a previous round of review and you feel that this manuscript is now acceptable for publication, you may indicate that here to bypass the “Comments to the Author” section, enter your conflict of interest statement in the “Confidential to Editor” section, and submit your "Accept" recommendation.

Reviewer #1: All comments have been addressed

Reviewer #2: All comments have been addressed

2. Is the manuscript technically sound, and do the data support the conclusions?

Reviewer #1: Yes

Reviewer #2: Yes

3. Has the statistical analysis been performed appropriately and rigorously? 

Reviewer #1: Yes

Reviewer #2: Yes

4. Have the authors made all data underlying the findings in their manuscript fully available?

Reviewer #1: Yes

Reviewer #2: Yes

5. Is the manuscript presented in an intelligible fashion and written in standard English?

Reviewer #1: Yes

Reviewer #2: Yes

6. Review Comments to the Author

Reviewer #1: No further comments, the authors have incooperated suggested earlier comments into their work that makes the work flow.

Reviewer #2: The added paragraph in the Discussion of this paper adequately addresses my previous concerns over the presentation of the results. I have no further concerns.

7. PLOS authors have the option to publish the peer review history of their article (what does this mean? ). If published, this will include your full peer review and any attached files.

**Do you want your identity to be public for this peer review?** For information about this choice, including consent withdrawal, please see our Privacy Policy .

Reviewer #1: **Yes: ** Dr Justus Simba MBChB, PhD.

Reviewer #2: No

---

## [Editor Report · Acceptance letter]

14 Jul 2023

PONE-D-23-00167R3 

Evaluation of CRP as a marker for bacterial infection and malaria in febrile children at the Douala Gyneco-Obstetric and Pediatric Hospital. 

Dear Dr. Ngwengi:

I'm pleased to inform you that your manuscript has been deemed suitable for publication in PLOS ONE. Congratulations! Your manuscript is now with our production department. 

If your institution or institutions have a press office, please let them know about your upcoming paper now to help maximize its impact. If they'll be preparing press materials, please inform our press team within the next 48 hours. Your manuscript will remain under strict press embargo until 2 pm Eastern Time on the date of publication. For more information please contact onepress@plos.org .

If we can help with anything else, please email us at plosone@plos.org . 

Kind regards, 

on behalf of

Dr. Marc Reismann 

Academic Editor

PLOS ONE